# Insertion Loss and Phase Compensation Using a Circular Slot Via-Hole in a Compact 5G Millimeter Wave (mmWave) Butler Matrix at 28 GHz

**DOI:** 10.3390/s22051850

**Published:** 2022-02-26

**Authors:** Noorlindawaty Md Jizat, Zubaida Yusoff, Azah Syafiah Mohd Marzuki, Norsiha Zainudin, Yoshihide Yamada

**Affiliations:** 1Faculty of Engineering, Multimedia University, Cyberjaya 63100, Selangor, Malaysia; noorlindawaty.jizat@mmu.edu.my; 2TM R&D Sdn Bhd, Cyberjaya 63000, Selangor, Malaysia; azahsyafiah@ieee.org; 3Communication Systems and Network (CSN i-Kohza) MJIIT, Universiti Teknologi Malaysia, Kuala Lumpur 54100, Johor, Malaysia; norsiha@siswa.um.edu.my

**Keywords:** via-hole, dual-layer, Butler matrix, compact size, low loss, millimeter wave, 5G

## Abstract

Fifth generation (5G) technology aims to provide high peak data rates, increased bandwidth, and supports a 1 millisecond roundtrip latency at millimeter wave (mmWave). However, higher frequency bands in mmWave comes with challenges including poor propagation characteristics and lossy structure. The beamforming Butler matrix (BM) is an alternative design intended to overcome these limitations by controlling the phase and amplitude of the signal, which reduces the path loss and penetration losses. At the mmWave, the wavelength becomes smaller, and the BM planar structure is intricate and faces issues of insertion losses and size due to the complexity. To address these issues, a dual-layer substrate is connected through the via, and the hybrids are arranged side by side. The dual-layer structure circumvents the crossover elements, while the strip line, hybrids, and via-hole are carefully designed on each BM element. The internal design of BM features a compact size and low-profile structure, with dimensions of 23.26 mm × 28.92 mm (2.17 *λ*_0_  ×  2.69 *λ*_0_), which is ideally suited for the 5G mmWave communication system. The designed BM measured results show return losses, *S_ii_* and *S_jj_*, of less than −10 dB, transmission amplitude of −8 ± 2 dB, and an acceptable range of output phase at 28 GHz.

## 1. Introduction

The growing demand for connectivity at high frequency bands is driving the expansion of millimeter wave (mmWave) radio frequency (RF) research for 5G technology. Compared with the previous generation’s 4G long term evolution (LTE) networks, 5G enables an increase in the peak data rate and allows for reduced power consumption, whilst also allowing for a 1 milli second roundtrip latency in real-time applications [1,2]. The higher frequency bands in mmWave allow higher capacity for the future 5G network [3]. However, the overall losses of mmWave systems are significantly larger than those of microwave systems for propagation characteristics.

Fortunately, the beamforming elements with a large-scale antenna array can be designed where larger gains can be produced, providing a robust communication link to enable mmWave communications (as the electrical length is reduced at the millimeter wave) [4,5]. Due to beamforming, there will be an increase in signal energy to the intended user and a decrease in the interference. The co-integration of the elements in the Butler matrix (BM) would not have been such a critical issue in a much lower GHz frequency band, as confirmed in [6,7]. As compared to the mmWave, it should be critically designed by controlling the inter-connection and inter-coupling among the BM elements, since, in the mmWave, the high path loss arises quadratically with the frequency (path loss ∝ f^2^) as defined by Friis’ law [8]. To combat the significant path loss and penetration loss at mmWave, the phase and amplitude of RF signals need to be accurately controlled.

Innovations on the mmWave beamforming using BM have been discussed in the previous reported references, using liquid crystal polymer, absorptive switch in a single planar, substrate integrated waveguide (SIW), and tri-layer configuration [9,10,11,12]. In [13], a planar BM is presented by removing the 0-dB crossover and in [14], a modified BM without phase shifter section is presented, which allows improvement of phase shift stability over a wide frequency range and reduces the length and the component loss. In [15], the broadband phase shifter is modified to obtain a mutual coupling reduction.

Some methods discussed in the literature achieve a low insertion loss but an increase in phase-error, while some methods achieve a low phase-error but otherwise suffer severe loss. The size of some of the Butler matrices, presented in the past literature, is impractical and their structure is complex (particularly when a larger matrix is needed), as the passive feeding network has *N* inputs and *N* outputs, where *N* is the power of 2 (*N* = 2*^n^*) [16]. A via-hole is used for the two-layer transition, and this technique can improve high-speed transmission [17,18]. The via-hole for the BM has also been discussed in [19], nonetheless it is designed at a lower frequency and has a transmission amplitude of more than −10 dB.

In this work, the novelty of the BM is from the via structure with a circular slot via-hole to connect the dual-layer substrate which provides minimum transmission amplitude, since the crossover is circumvented. The via-hole pin diameter and the slot diameter of the via-hole are used to control the coupling and the phase transition of the signal, where this enables the BM to minimize the imbalance on amplitude and phase. Thorough analysis of the transmission loss and phase delay shows the value of 0.029 dB/mm and 47°/mm for every change of the stripline. In the analysis of the via-hole, the structure with circular slot diameter, *D_S_* = 1.2 mm, via-hole pin diameter, *D_P_* = 0.3 mm, and substrate thickness, *D_T_* = 0.254 mm, provide minimal loss. The compact configuration of dual-layer makes the BM size suitable for the actual application at 28 GHz with limited space. The analysis of the extension of strip line used is performed and provides the return losses, *S_ii_* and *S_jj_*, of less than −10 dB, transmission amplitude of −8 ± 2 dB, and an acceptable range of output phase at 28 GHz.

## 2. Strip Line and Via-Hole Parameter Analysis

The dielectric material considerations include the dielectric constant and loss tangent. The imaginary component of the loss factor is related to the degree of incident electromagnetic (EM) energy losses in the material, whereas the real part is related to the ability of a material to store the EM energy through wave propagation. It is also known as the dissipation factor that describes the angle difference between capacitance current and voltage. A lower loss tangent is needed to maintain low dielectric loss and low dielectric absorption. The loss tangent, *tan δ*, measures the signal loss as the signal propagates through the transmission line, expressed in (1) [20]. These two parameters, *ɛ_r_* and *tan δ*, are directly related to the *Q*-factor, due to the dielectric, *Q_D_*, which can be expressed as (2).
(1)$$tan\partial =\frac{\omega \varepsilon_{r}\prime \prime +\sigma }{\omega \varepsilon_{r\prime }}\;$$
(2)$$Q_{D}=27.3\frac{\sqrt{\varepsilon_{eff}}}{\alpha_{d}\lambda_{0}}$$
where *ε_eff_*, *λ*_0_, and *α_d_* are the effective dielectric constant, the wavelength in the air, and dielectric loss.

A high *Q*-factor is preferred since it gives less loss and resistance. Since, in most applications, a high *Q*-factor component is needed, these two material parameters need to be carefully chosen to avoid the skin effect and inter-winding capacitance that can influence the resistance and reactance at higher frequencies. The via is smaller than the working wavelength and is modelled as lumped elements. The capacitance of the via is caused by the electric field that exists between the via and the ground layers on the board, while the inductance is caused by the magnetic field that exists around the portion of the via that carries the signal current [21]. In [22,23,24,25,26], the capacitance, *C*, and the inductance, *L*, equivalent lumped element configuration is shown in Figure 1. The dimension of the circular hole between the via conductor and the ground plane determine the capacitance and inductance. Every via has parasitic capacitance to the ground. If the ground clearance is too small, the via will appear capacitive; if it is too large, the via will be inductive [27].

The via has parasitic capacitance, as discussed in [28,29]. The capacitance associated with the via is obtained from Gauss’s law of coaxial capacitor formula, in which the field magnitude equals the absolute value of the total charge enclosed, divided by the surface area in (3) and (4). The simplified capacitance is expressed in (5) [30]:(3)$$E=\frac{Q}{2\pi \varepsilon rL}\hat{r}$$
(4)$$V=-\int_{a}^{b}\frac{Q}{2\pi \varepsilon rL}\hat{r}\cdot \hat{r}dr$$
(5)$$C=\frac{1.41\varepsilon_{r}TD_{P}}{D_{S}-D_{P}}$$
where *D_S_* represents the diameter of the clearance circular hole in the ground plane, *D_p_* is the diameter of the via-hole pin, *h* is the thickness of the substrate, and $\varepsilon_{r}$ is the relative electric permittivity. All dimensions are expressed in inches.

Figure 2 shows the via model through the ground plane perspective, and Figure 3 shows the analysis of the via-hole as a function of the thickness of the substrate, *T (h*_1_
*+ h*_2_*)*, and circular slot diameter, *Ds*, towards the value of the capacitance for a via with a fix value of via-hole pin diameter, *D_p_,* of 0.3 mm. The results show the thickness, *T*, and the circular slot diameter, *Ds* (dimensions in inches), variation affect the value of the via-hole capacitance. The capacitance increases sharply when the ground plane is nearer to the microstrip via.

The magnetic line of force is created due to the signal current to the concentric circles centered on the signal via. The field strength is parallel to the via, where the field reduces when the distance increases from the via. The current flowing along the via provides a partial inductance, *L*. The via behaves as a parasitic series inductance as in (6),
(6)$$L=0.508h\left[ln\frac{4h}{d}\right]+1\;$$
where *L* represents the inductance of the via in nH, *h* is the length of the via and *d* is the diameter of the via (both expressed in inches).

Figure 4 shows the analysis of the via-hole as a function of via length, *h*, and via-hole pin diameter, *D_p_*, towards the value of the via inductance. The results show the via length, *h*, and the via-hole pin diameter, *D_p_* (dimensions in inches), variation effect the value of the via-hole inductance. The inductance increases sharply when the microstrip via-hole is smaller.

The strip line is analyzed using the NPC-F220A substrate with thickness of 0.254 mm, dielectric constant of 2.2 and tangent loss of 0.0007, and the *Q* factor is 554.7 from Equation (2). The strip line length, *L*_0_, and *W*_0_, are 7.75 mm and 0.84 mm, obtained from the computer simulation technology (CST) simulation by considering the impedance of 50 ohms and electrical length of 360 degrees at operating frequency of 28 GHz. Figure 5 shows the strip line configuration and the current distribution. The guided wavelength, *λ_g_*, is 7.2 mm, from (7)–(8) with a total length of 360°. Thus, a 0.02 mm line length produces 1° of phase delay. The additional strip-line, Δ*L*, can affect the insertion loss and phase shift. In this study, the length change, Δ*L*, is varied from 1 mm to 10 mm. The insertion loss and phase shift characteristic from the simulation can be seen in Figure 6, with 0.029 dB/mm and 47°/mm changes for every strip line dimension.
(7)$$\lambda_{0}=\frac{c}{f}\;$$
(8)$$\lambda_{g}=\frac{\lambda_{0}}{\sqrt{\varepsilon_{r}\mu_{r}}}$$


The via-hole produces a minimal loss at high frequencies by changing the parameter of the ground circular slot diameter, *D_S_*, via-hole pin diameter, *D_P_*, and the substrate thickness, *D_T_*. The via-hole pin has greater flexibility when it connects the dual-layer substrates, as shown in Figure 7. The strip line width, *w*, and length, *l*, are 0.7826 mm and 4 mm. An analysis with the different circular slot diameter, *D_S_*, the via-hole pin diameter, *D_P_*, and substrate thickness, *D_T_*, (Figure 8) towards the insertion loss and the output phase, showed that minimized insertion loss is obtained when *D_S_* is 1.2 mm, *D_P_* is 0.3 mm and *D_T_* is 0.254 mm by considering the other two parameters are constant for every analysis. Significant changes of output phase are observed when the circular slot diameter, *D_S_*, and the via-hole pin diameter, *D_P_*, are changed.

## 3. Butler Matrix

The BM is a passive network with the selection of *N* inputs and *N* outputs, where *N* is the power of 2 (*N* = 2*^n^*). When any of the input ports is excited, the output ports have equal amplitudes and progressive output phases. The substrate used to design the BM is NPC-F220A from Nippon Pillar Packing Co. Ltd., Osaka, Japan with thickness of 0.254 mm, dielectric constant of 2.2 and tangent loss of 0.0007. This substrate offers the superior dielectric properties of fluorocarbons resin film for high frequency waveband, specially to equip the multi-layer board application. The conventional design of 4 × 4 planar BM has four hybrids and two crossover and four phase shifters (as illustrated in Figure 9a).

Due to its high complexity, this project proposed a new method by circumventing the crossover structure. Implementing the dual-layer structure with a via-hole to connect between these two layers avoids crossover connections, which reduces the size of the BM and reduces losses. The circular slot is designed at the common ground at the centre between these two layers to improve the performance of the insertion loss and the phase transition. The structure is compact, where the two sides of the hybrid are stacked side by side (Hybrid 1 and Hybrid 2) with Hybrid 1 at the upper layer and Hybrid 2 at the lower layer. This is applicable to Hybrid 3 and Hybrid 4, as shown in Figure 9b. The new configuration is simplified, where the crossover and the phase shifter are circumvented in the BM design.

The via-hole compensates the phase transition, *ϕ*_1_ and *ϕ*_2_, in the BM, when simulated with the Reference ‘*A*’ strip line. The signal current flow is illustrated in Figure 10, when the input is fed with signals. The input signal will be equally divided when it passes through the hybrid. When the signal is passing through node #via1 and #via2, the signal will be transitioned to the other substrate layer.

In Figure 10a, if the signal is fed to P1, the signal passes node A (Hybrid 1) and transitions to the lower layer (#via 1). The signal passes node B to node C, P5. The other signal is coupled in Hybrid 2 to node D, P7. The signal at node A (Hybrid 1) and node E is coupled (Hybrid 4) to P6 while the other signal from node A is coupled in Hybrid 1 and Hybrid 4 to P8. The other signal flow is shown in Figure 10b–d, when the input is fed to the input ports P2, P3, and P4, respectively. The Butler matrix is a passive feeding network with *N* inputs and *N* outputs, where *N* is the power of 2 (*N* = $2^{n})$. For the *p^th^* beam angle, the phase differences between the output ports are given by (9) with the phase difference between output ports are ±45° and ±135° [31]:
(9)$$\phi_{P}=\pm \frac{2p-1}{N}\times 180^{{}^{\circ}}$$
where *N* = 4, *n* = 2 and *p* = 1, 2, …. (*n* + 1).

The signal flows through each BM element with equal power at different phase [32]. The resulting signal at each output port has the same output phase difference. Accurate output phases are achieved by considering the 0° phase delay for every path of the hybrids, the controlled strip line, and the via-hole pins, *ϕ*_1_ and *ϕ*_2_. To achieve an identical phase difference for the sequenced ports, Equations (10)–(13) need to be satisfied.
(10)$$∡S_{81}-∡S_{71}=∡S_{71}-∡S_{61}=S_{61}-∡S_{51}=-45{}^{\circ}\;\;$$
(11)$$∡S_{82}-∡S_{72}=∡S_{72}-∡S_{62}=∡S_{62}-∡S_{52}=+135{}^{\circ}$$
(12)$$∡S_{83}-∡S_{73}=∡S_{73}-∡S_{63}=∡S_{63}-∡S_{53}=-135{}^{\circ}$$
(13)$$∡S_{84}-∡S_{74}=∡S_{74}-∡S_{64}=∡S_{64}-∡S_{54}=+45{}^{\circ}$$

### 3.1. 3-dB Hybrid

The 3-dB hybrid is achieved by two transmission lines of *λ*/4 vertical and horizontal branches. The horizontal and vertical branch of the hybrid is optimized to control the minimum coupling value and the progressive output phase at the output ports. S-parameters relate incident and reflected voltage waves at the network’s various ports. This is defined in (14) [33]:(14)$$\left[V^{-}\right]=\left[S\right]\left[V^{+}\right]$$
where $\left[V^{-}\right]\;$is the reflected voltage wave vector, $\left[V^{+}\right]$ is the incident voltage wave vector, and S is the scattering matrix. The elements of these matrices refer to the magnitude and phase.

Figure 11 shows a hybrid with four input ports (#1, #2, #3, #4), where the signal becomes transmission when the signal flows from #1 to #2, and the coupling properties when the signal flows from #1 to #3. In the proposed design, the hybrid has a dimension of 4.66 × 6.00 mm^2^. The hybrid has width, *W_zo_*, as the feeder width and *W_z_* is the width of the vertical branch with impedance *Z_0_*, 50 ohms, and W_z1_ is the width of the horizontal branch with impedance of *Z_0_*/√2, 35.35 ohms. *L_zo_* is vertical length branch, almost a quarter wavelength (*λ*_0_/4) and L_z_ is horizontal length branch, where *L_z_* is longer than *L_z0_* in order to produce a 90-degree phase difference at the output ports.

When the input signal is fed to input 1, the signal is evenly distributed to output 2 and 3. No power is delivered to the isolation port 4. Similarly, when an incident wave is fed into input port 4, power couples to ports 2 and 3, while port 1 is isolated due to its symmetrical properties. In the hybrid, the coupling (*S_31_*) have the value of −3 dB, which represents the power signal is equally divided to each output ports. The return loss (*S_11_*) and the isolation (*S_41_*) provide values greater than −10 dB, and the phase output difference is 90°.

The distribution of amplitudes to each output port is shown in Figure 12a, with *S_21_* and *S_31_* having values of −3 ± 1 dB at 28 GHz. Return loss and isolations of *S_11_* and *S_41_* show values less than −20 dB, which provides the higher percentage of signal transmission. In Figure 12b, the simulated output phases between output port 2 and 3, produce a phase difference of 90 ± 5°, where the phase between these output ports is characterized by Angle (*ϕ_31_*) − Angle (*ϕ_21_*). Detailed dimensions of the hybrid and finalized phase difference between output ports are tabulated in Table 1.

### 3.2. Via-Hole

The via-hole BM design is simulated using the computer simulation technology (CST) microwave studio software. The design has two input ports (Port 1, Port 2) and two output ports (Port 6, Port 8) on the top layer, while the other four ports are on the lower layer, with Port 3 and Port 4 as input ports and Port 5 and Port 7 as output ports. The via-hole pin connects the top layer to the bottom layer and provides a path for an electrical signal through it. The E-field indication through the circular slot hole is illustrated in Figure 13 and the surface current distribution from one layer to another using via-hole pin is illustrated in Figure 14.

The dual-layer design circumvents crossover structure, where the strip line of reference ‘*A*’ that is shown in Figure 15 is used to connect between the hybrids. The phase shift between the via-hole and reference ‘*A’* is controlled until a *ϕ*_1_ = *ϕ*_2_ = 45° phase difference. Table 2 shows the detailed dimension of the configuration and the phase shift between the two outputs is detailed in Figure 15. As illustrated in the BM internal design (top view) in Figure 16a, the hybrid is stacked side by side at the different substrate and the BM structure is compact with the dimensions of 23.26 mm × 28.92 mm. The via-hole connects between the upper substrate and the lower substrate for electrical path transition, with 45° phase difference with the Reference ‘*A*’ where the current distribution is illustrated in Figure 16b. The phase difference is calculated by *S*_43_ (Phase) − *S*_21_ (Phase) = *ϕ*_1_ = *ϕ*_2_
*=* 45°.

The return loss, *S_ii_* and *S_jj_*_,_ for input ports, *i*, and output ports, *j*, are below −10 dB, as shown in Figure 17. The transmission amplitude, *S_ij_*, shows coupling of 6 ± 2.2 dB with minimum loss of 0.1 dB in the range of −6.1 dB to −8.2 dB, as shown in Figure 18. Figure 19 shows the theoretical and simulation results of the output phase difference for each BM output ports when the input signals are fed, where the theoretical value is calculated based on Equation (9) [31]. The phase shift is critically controlled, based on the dimension of reference ‘*A*’, strip line and at the connection between every hybrid, according to the designed electrical length.

Table 3 details the simulated transmission amplitude of the via-hole BM with balanced values with minimal average difference of 1.1 dB from the ideal value of −6 dB, where the ideal value for transmission amplitude of −6 dB is assumed lossless for the structure. The output phase difference of the theoretical and simulation is tabulated in Table 4, and the simulation results agree well with the theoretical value and have average errors of 4.7° from the ideal values. The measured 10 dB impedance bandwidth of the structure ranges from 26 to 30 GHz, which is equivalent to 14% of bandwidth. The size of the structure is 2.17 *λ*_0_ × 2.69 *λ*_0_ at the center frequency of 28 GHz. The internal structure of the BM and the new structure with extension strip line, Δ*L*, with input ports (1–4) and output ports (5–8) is illustrated in Figure 20. The final design of BM that includes the additional strip line has a new dimension of 100 mm × 82.5 mm, equivalent to 9.34 *λ*_0_ × 7.7 *λ*_0_ at the center frequency of 28 GHz.

## 4. Results and Discussion

The BM is fabricated using the NPC-F220A substrate with thickness of 0.254 mm, dielectric constant of 2.20 and thermal coefficient of 0.0007. The calibration of the return loss is performed using one cable from the Keysight PNA N 5224A Vector Network Analyzer (VNA) to each of the input and output ports of the BM (one-port measurement). To perform the transmission amplitude calibration, (two-port measurements), the length of the transmission cable reference is assembled, based on the total length of the designed current path of the BM, where the total electrical length, ∑*L*, is calculated from the starting point of the input port to the output port of the BM (as shown in Figure 21).

Initially, the measurement connection to the internal compact design of the BM (the size of the structure is 23.26 mm × 28.92 mm) is performed as shown in Figure 22. Due to a very small spacing between the ports of the BM internal design and the VNA cable, the board is quite unstable during the measurement. Thus, some adjustments are made with the reference cable of 100 mm with the new structure of the BM with feeding strip line extension of Δ*L =* 30 mm as shown in Figure 23. The adjustment is made by considering the analysis of the extension strip line, as discussed earlier in Figure 7. The final design of BM, which includes the additional strip line, has a new dimension of 100 mm × 82.5 mm. The scattering parameter for the return loss and transmission amplitude is measured using the two-port VNA. During the via-hole assembly, the two substrates need to be accurately aligned to each other, so the signal transition between the layers can be controlled and the ground placed properly. The via-hole pin is also covered with insulator paint to avoid a short circuit.

Both measurement results of the return loss for each input and output port, *S_ii_* and *S_jj_*, show values below the reference point, −10 dB, as shown in Figure 24. The values below −10 dB show that the BM structure has good return loss. It can be observed that the measured results are slightly changed compared to the simulation. The frequency shifting might be attributed to the fabrication tolerance and variations in the high frequency connector properties. The structure is very small at the millimeter wave. This affects the final design of the Butler matrix, especially when the output is connected to the millimeter wave connector. In the design, there are four input connectors and four output connectors. In order to get a stable structure and promising measurement, the extension of strip line is used to satisfy the actual connector spacing. Moreover, the measured results observed are comparable and provide values below the reference point.

The output power to input power ratio is represented by the transmission amplitude with the theoretical value of −6 dB. For improved comparison, the simulation and measurement values of transmission amplitude are plotted in Figure 25. The amplitude simulation with the extended line shows the range from −7.2 dB to −9.3 dB, while the measurement shows the range from −8.2 dB to −10 dB. The imbalanced transmission amplitude measurement has an average deviation of 1.38 dB compared to the simulation values. The variation of the transmission amplitude from the ideal value of −6 dB is due to the extension of the additional strip line. The strip line loss (referring to Figure 6) has line loss offset of 0.299 dB for a line length of 10 mm. From the simulation, the estimated strip line loss is 0.9 dB for an extension line length, Δ*L*, of 30 mm. In comparison, the addition of ±0.5 dB to some of the ports might be attributed to the fabrication tolerance and variations in the high frequency connector properties.

The simulation and measurement output phase difference results are shown in Figure 26. The output phase difference might be due to the extended strip line loss as discussed in the previous analysis (referring to Figure 6), where the phase delay is −47°/mm resulting in *ϕ*Δ*L* of −30° for Δ*L* = 30 mm. Although the phase delay is shifted with an average of 32° compared to the simulation, the trend of the output phase difference for each respective input can be seen to be progressive. The progressive phase difference at each respective output port is comparable to the theoretical value of ±45° and ±135° considering phase loss due to the extension line.

To establish the significance of the proposed method, Table 5 summarizes the comparison between the proposed BM and the previous works reported in the literature. 

All of the proposed methods were carefully investigated with the co-integrated elements in the BM design. Nguyen [34] reported a minimum transmission amplitude from −8 to −10 dB, where the BM is designed based on a dual-layer structure through the Finline method. A compact design is reported using CMOS [35]. CMOS technology has been proven as an effective platform for low power and small size mm-wave transceiver systems. However, the CMOS designs have high complexity and mostly used low resistivity substrate and high sheet resistance. Other reported mmWave BM designs use substrate integrated waveguide and open-short stubs. However, these designs achieved moderate transmission amplitude [36,37]. In [38], the BM is designed by milling and screwing together nine aluminum plates. Although the insertion loss is small (around 1.8 dB), the fabrication requires quite a complicated process.

This work is also compared with the conventional planar BM at 28 GHz. One of the methods used a single-pole four-throw (SP4T) absorptive switch in a single planar printed circuit board, as discussed in [39]. In [40], the BM is designed using planar configuration which uses the conventional crossover and phase shifter with insertion loss from 5.3 to 10 dB. Due to the design complexity, each planar design has a size of 3.45*λ*_0_ × 4.7*λ*_0_ and 3.4*λ*_0_ × 3.4*λ*_0_, respectively. In the proposed design, the BM structure uses a dual-layer substrate, and circumvents the usage of the crossover while having the via-hole to control the insertion loss and the output phase difference. The design is simpler, cost-effective, and easy to fabricate, and therefore can be easily modified for the higher order configuration of matrix for more port’s BM requirements.

## 5. Conclusions

In this project, a compact dual-layer BM is designed where two sides of the hybrid are stacked side by side in the different substrate. The via-hole pin is used to connect between the two layers and as a medium for signal transition. The variation of via-holes with different dimensions of the circular slot diameter, *D_S_*, via-hole pin diameter, *D_P_*, and substrate thickness, *D_T_*, towards the insertion loss and phase delay is analysed. The size of the internal structure is compact with 2.17*λ*_0_ × 2.69*λ*_0_ at the center frequency of 28 GHz. The dual-layer and via-hole circumvents the crossover structure in BM, resulting in a lossless design, where the transmission amplitude of the internal design of BM is −6.1 dB to −8.2 dB, with average deviation of 1.1 dB and 4.7° from the ideal values. The new structure of the BM, with feeding strip line extension of Δ*L* = 30 mm, is considered due to the measurement limitation. The measured results showed return losses of less than −10 dB for both the input and output ports at 28 GHz. The strip line extension is added to the internal design to satisfy the fabrication and measurement process. The strip line loss analysis has line loss offset of 0.029 dB/mm and −47°/mm phase delay. The proposed BM features a transmission amplitude of −8 ± 2 dB with average deviation of 1.38 dB, and average deviation of 32° for phase, compared to the simulation results. The measured prototype achieved the comparable output phase difference from the ideal values of ±45° and ±135° by considering the analysis of the phase delay due to the strip line extension. The validated result can be further extended into the beamforming system and can be integrated into a larger matrix for 5G wireless communication with antenna array at 28 GHz.

## Figures and Tables

**Figure 1 sensors-22-01850-f001:**
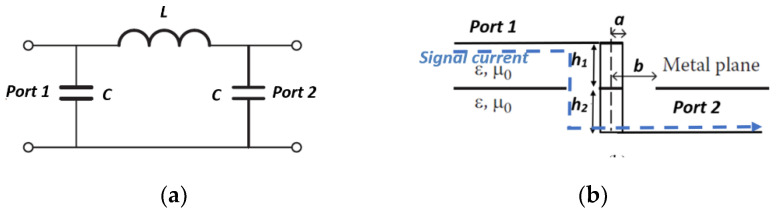
Via (**a**) equivalent circuit (**b**) cross section; where *a* represents the radius of the via-hole pin, *b* represents the radius of the circular slot *h*_1_ + *h*_2_ is the thickness of the via-hole pin.

**Figure 2 sensors-22-01850-f002:**
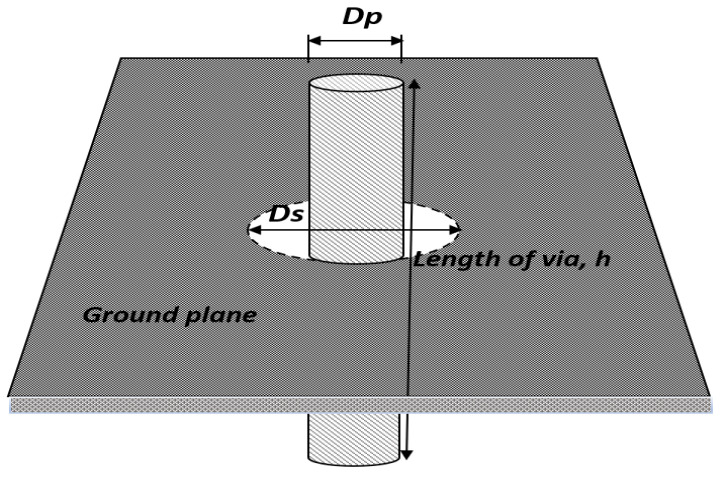
Via model through ground plane perspective.

**Figure 3 sensors-22-01850-f003:**
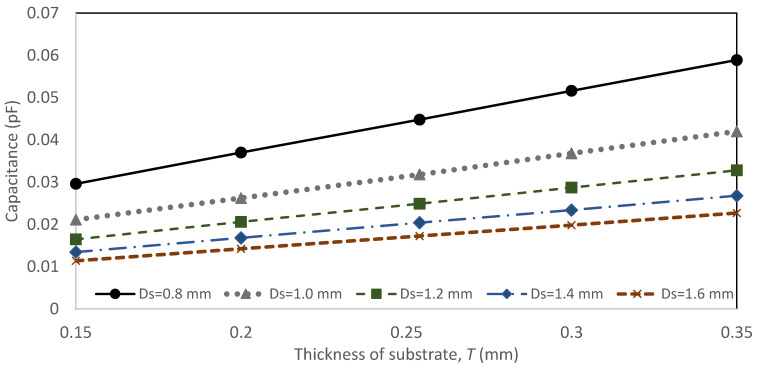
Via hole capacitance as a function of thickness of substrate, *T* and different diameter of the circular slot, *Ds*.

**Figure 4 sensors-22-01850-f004:**
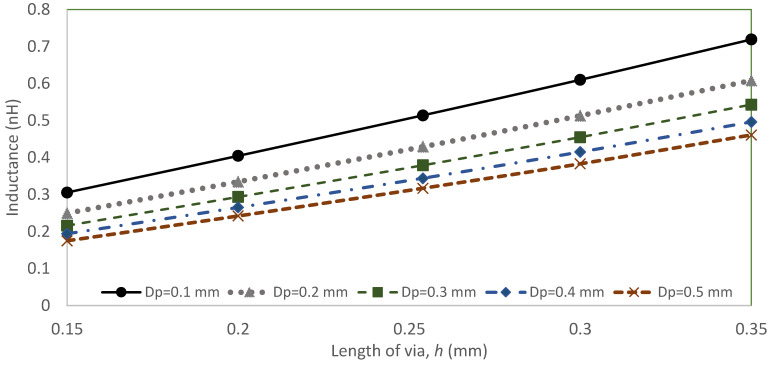
Via hole inductance as a function of length of via, *h* and different diameter of the microstrip via, *D_p_*.

**Figure 5 sensors-22-01850-f005:**
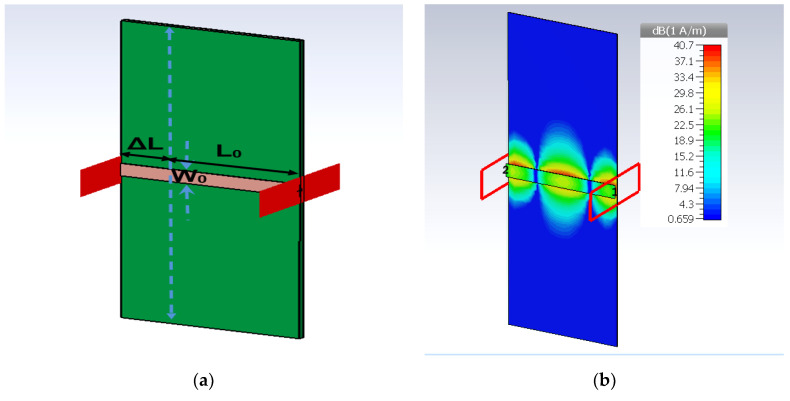
Configuration of (**a**) strip line and (**b**) current distributions.

**Figure 6 sensors-22-01850-f006:**
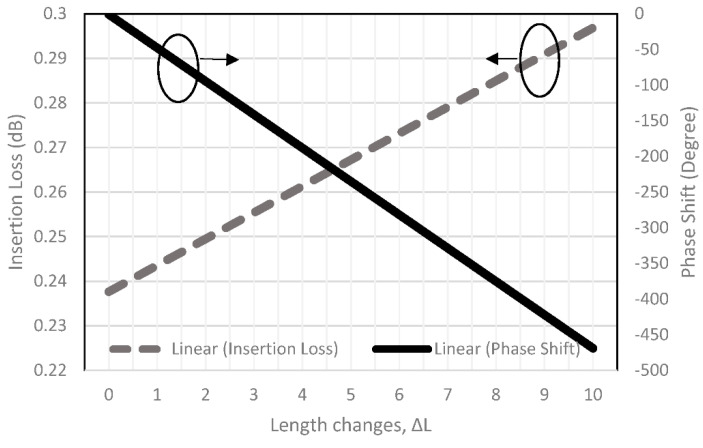
Insertion loss and phase shift of the strip line length changes, Δ*L*.

**Figure 7 sensors-22-01850-f007:**
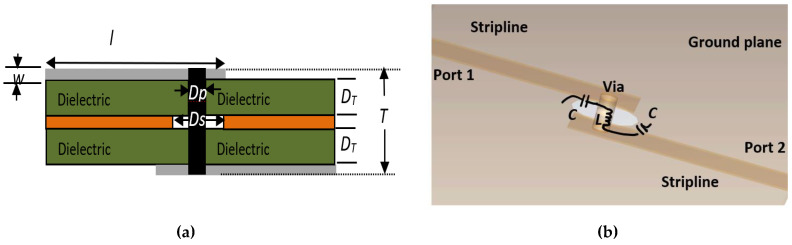
Configuration of via-hole with strip line (**a**) configuration and (**b**) simulation design.

**Figure 8 sensors-22-01850-f008:**
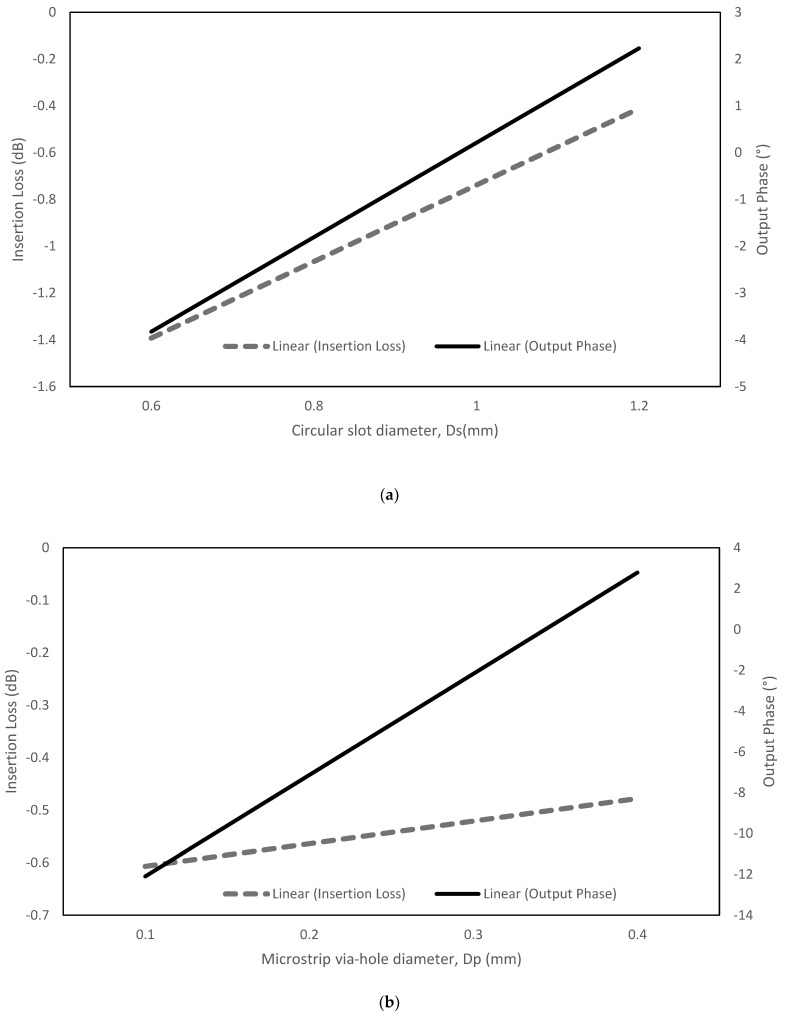
Parameter analysis towards the insertion loss: (**a**) circular slot diameter, *Ds*; (**b**) via-hole pin diameter, *D_P_*; and (**c**) substrate thickness, *D_T_*.

**Figure 9 sensors-22-01850-f009:**
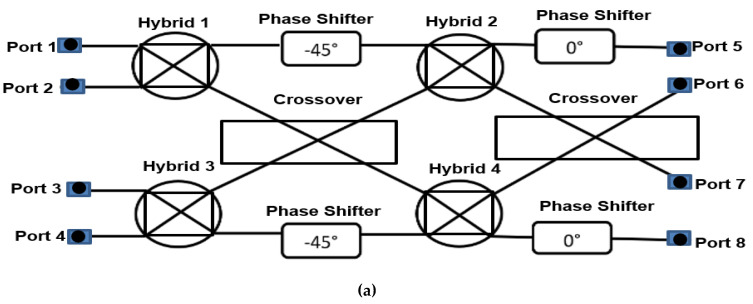
Butler matrix structure: (**a**) conventional and (**b**) using a via-hole.

**Figure 10 sensors-22-01850-f010:**
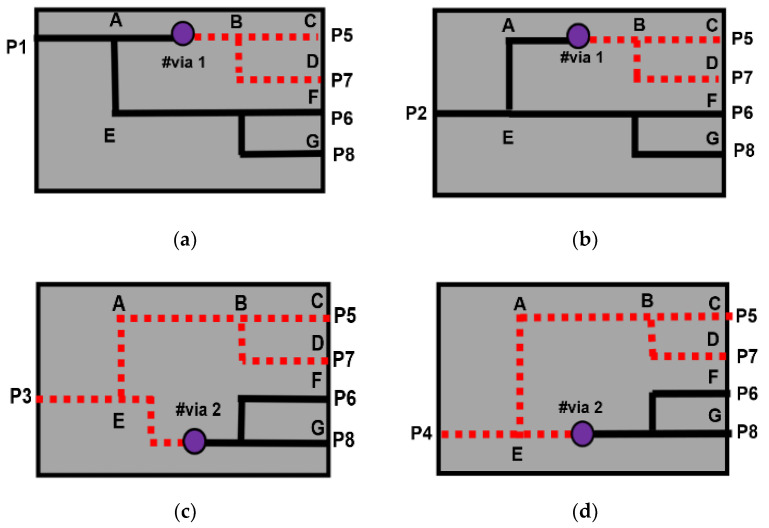
Current signal flow when input fed to: (**a**) port 1, (**b**) port 2, (**c**) port 3, and (**d**) port 4.

**Figure 11 sensors-22-01850-f011:**
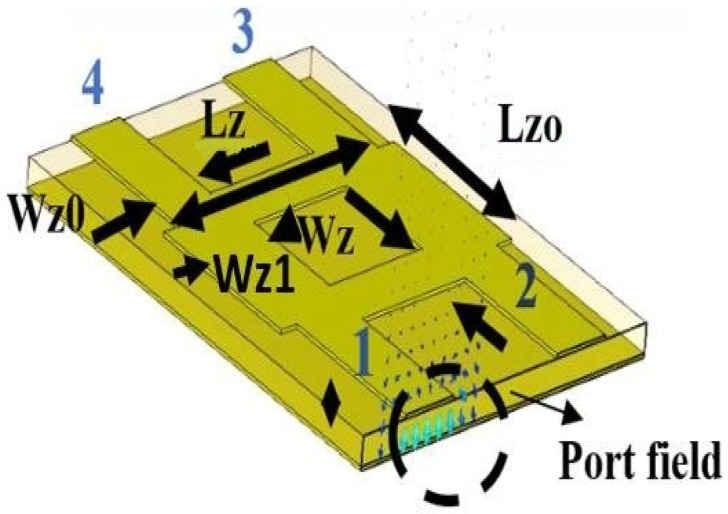
Hybrid structure topology dimensions.

**Figure 12 sensors-22-01850-f012:**
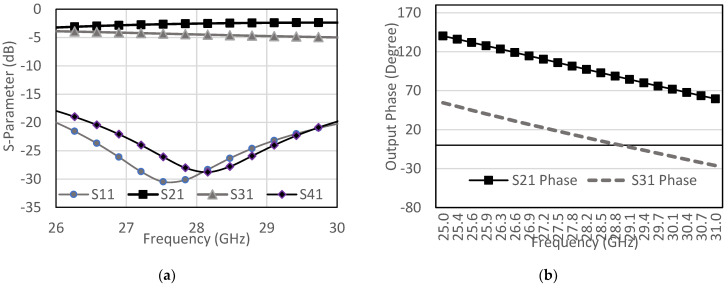
Hybrid performance: (**a**) *S*-parameter and (**b**) phase at the output ports.

**Figure 13 sensors-22-01850-f013:**
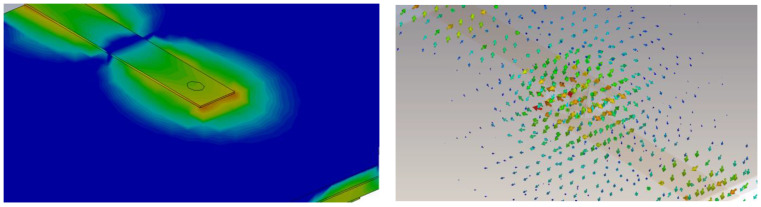
E-field indication through via-hole pin current density.

**Figure 14 sensors-22-01850-f014:**
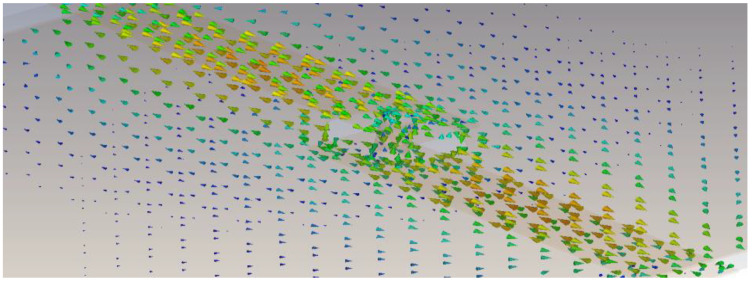
Surface current distribution from one layer to another using via-hole pin.

**Figure 15 sensors-22-01850-f015:**
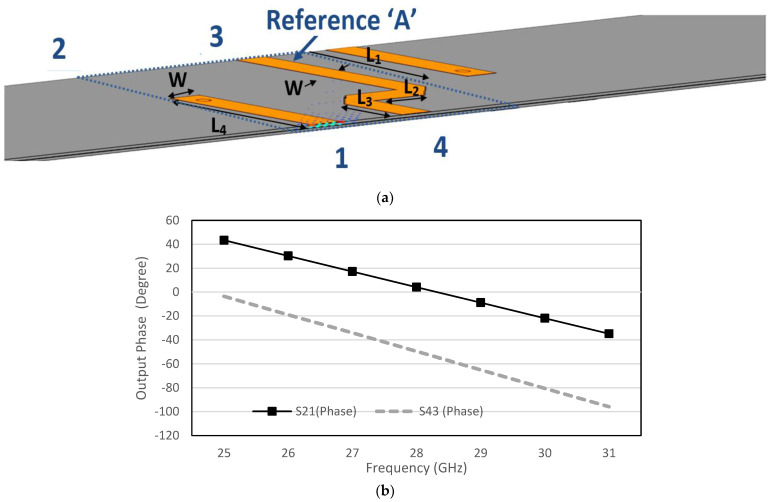
Via-hole with reference ‘*A*’: (**a**) configuration and (**b**) output phase, *ϕ*_1_ and *ϕ*_2_.

**Figure 16 sensors-22-01850-f016:**
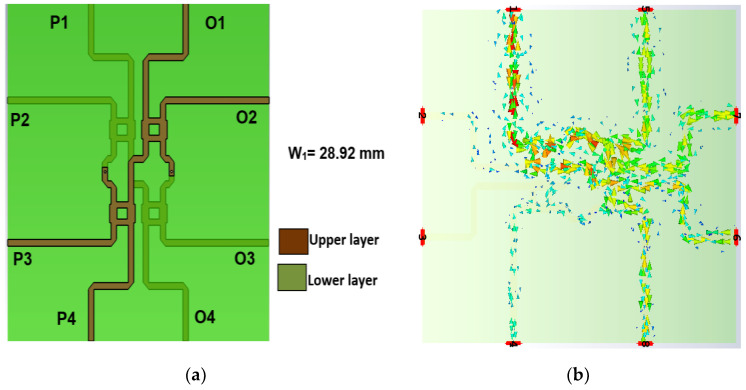
Butler matrix internal design: (**a**) top view and (**b**) surface current distribution.

**Figure 17 sensors-22-01850-f017:**
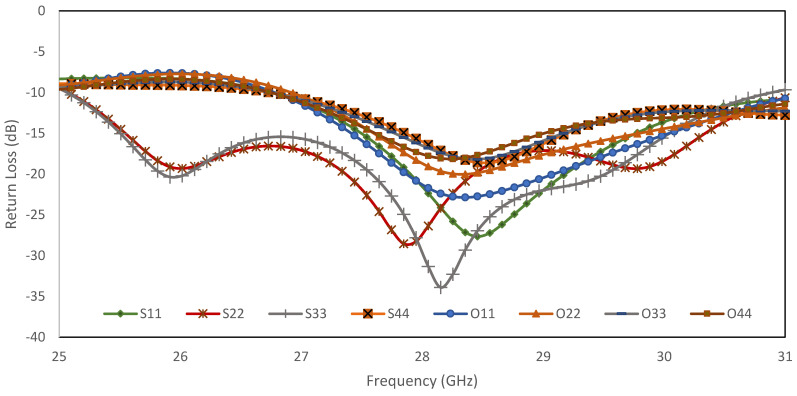
Return loss simulation of the proposed BM for input ports (S11–S44) and output ports (O11–O44).

**Figure 18 sensors-22-01850-f018:**
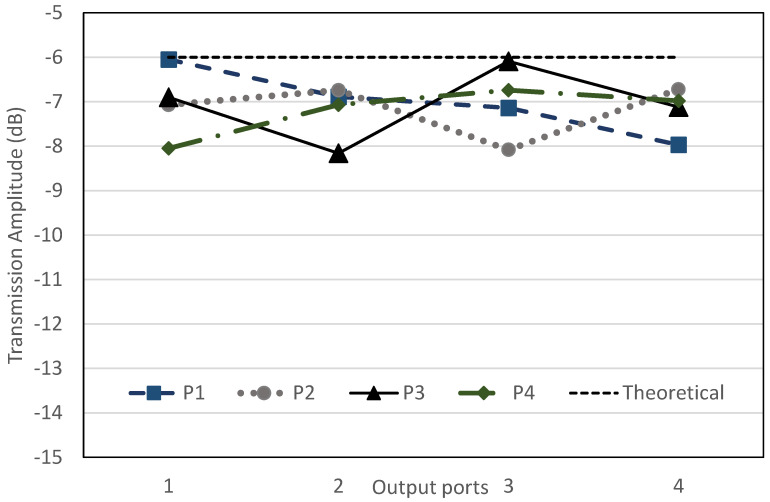
Transmission amplitude, *S_ij_*, of the proposed BM.

**Figure 19 sensors-22-01850-f019:**
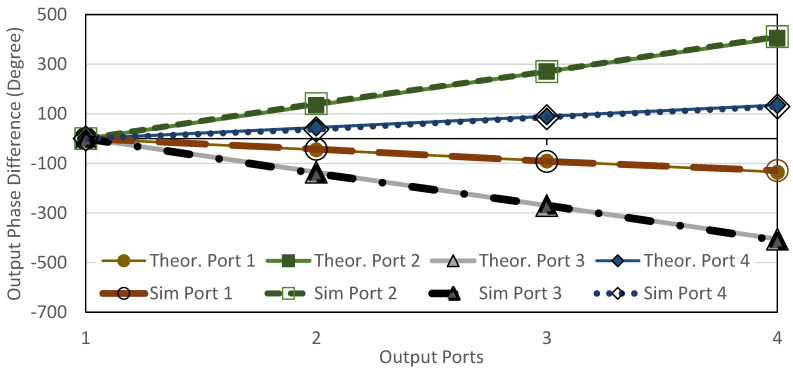
Output phase difference of the BM.

**Figure 20 sensors-22-01850-f020:**
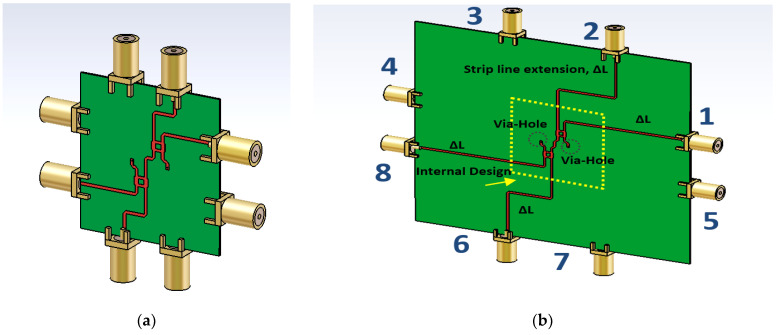
BM structure with (**a**) internal design (**b**) internal design with strip line extension, Δ*L*.

**Figure 21 sensors-22-01850-f021:**
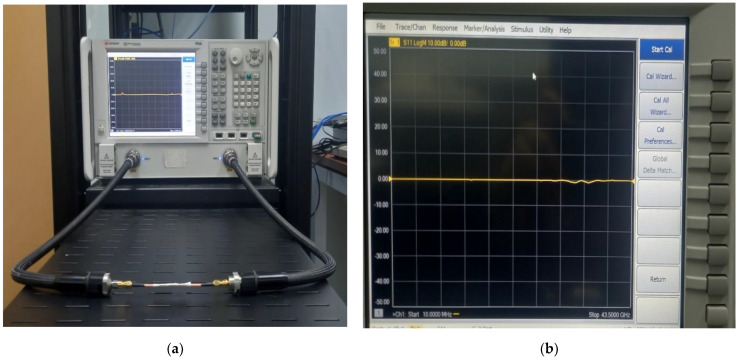
Transmission calibration configuration for two ports: (**a**) reference cable and (**b**) reference value.

**Figure 22 sensors-22-01850-f022:**
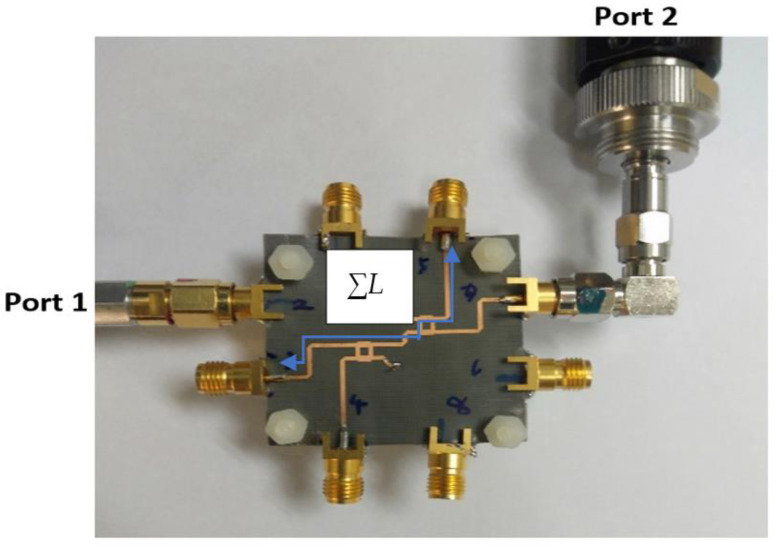
Transmission measurement of internal design from input to output port.

**Figure 23 sensors-22-01850-f023:**
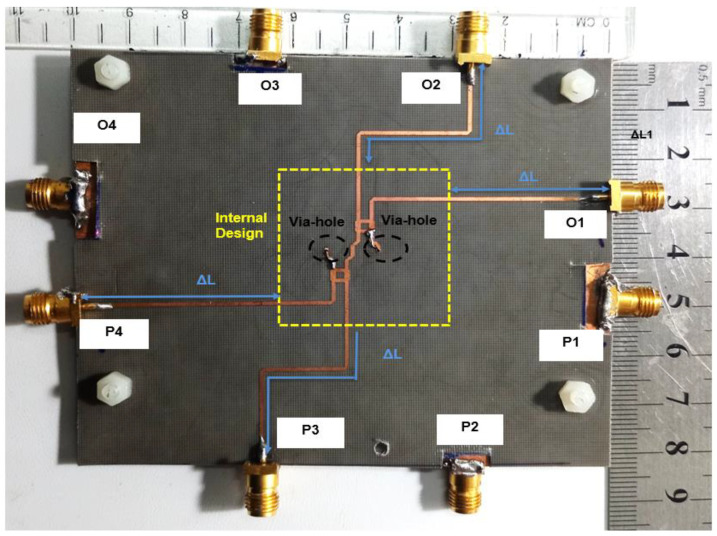
Butler matrix prototype with the extended line length, Δ*L*.

**Figure 24 sensors-22-01850-f024:**
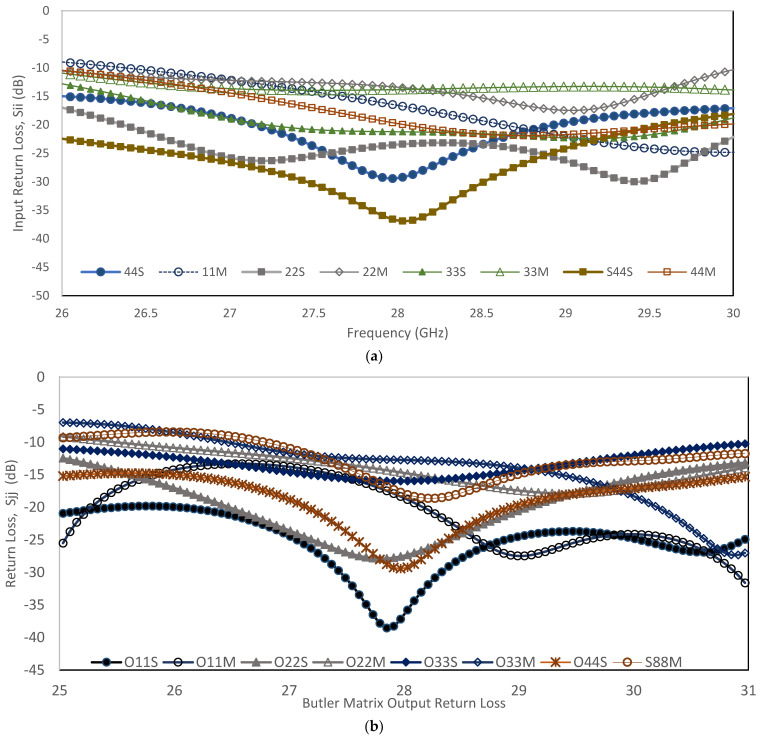
Return loss simulation and measurement results: (**a**) input ports, P1 till P4; (**b**) output ports, O1 till O4.

**Figure 25 sensors-22-01850-f025:**
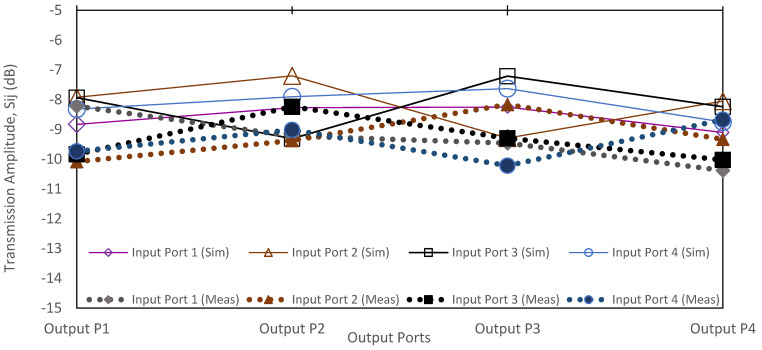
The transmission amplitude simulation and measurement.

**Figure 26 sensors-22-01850-f026:**
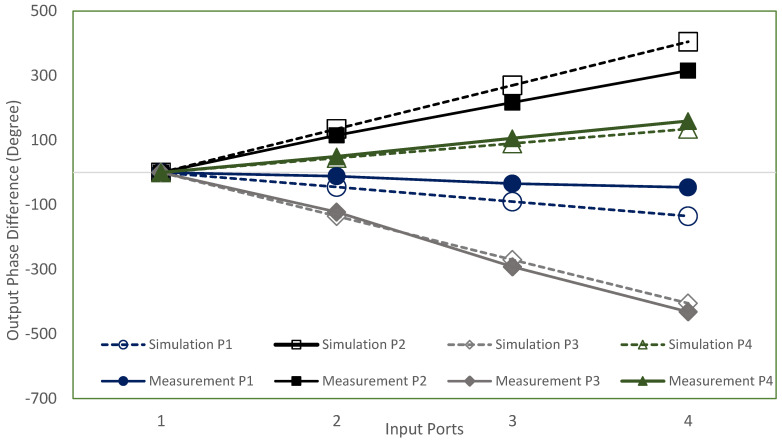
The output phase simulation and measurement.

**Table 1 sensors-22-01850-t001:** Hybrid dimensions and output phase.

Hybrid Dimension (mm)					Output Phase (°)	
*L_z_*	*L_zo_*	*W_z_*	*W_z0_*	*W_z1_*	*S_31_*	*S_21_*
3.14	2.11	0.68	0.68	1.22	99.5	10.47

**Table 2 sensors-22-01850-t002:** Via-hole with reference ‘*A*’ dimensions and output phases.

Hybrid Dimension (mm)					Output Phase (°)	
*L* _1_	*L* _2_	*L* _3_	*L* _4_	*W*	*S* _43_	*S* _21_
4.56	1.40	1.64	4.01	0.78	49.69	4.17

**Table 3 sensors-22-01850-t003:** Simulated transmission amplitude of the via-hole BM in dB.

Transmission Amplitude of the Via-Hole BM (Internal Design) (dB)				
	P1	P2	P3	P4
1	−6.1	−6.9	−7.1	−8.0
2	−7.1	−6.7	−8.0	−6.7
3	−6.9	−8.2	−6.1	−7.1
4	−8.1	−7.1	−6.7	−6.9

**Table 4 sensors-22-01850-t004:** Theoretical and simulation of output phase difference.

Theoretical Phase (°)					Simulated Phase (°)			
	P1	P2	P3	P4	P1	P2	P3	P4
1	0	−45	−90	−135	0	−42.3	−91.9	−129
2	0	135	270	405	0	142	271	411
3	0	−135	−270	−405	0	−141	−269	−410
4	0	45	90	135	0	37	87	128

**Table 5 sensors-22-01850-t005:** A comparison with existing mmWave BM.

Ref.	Size Dimensions	Freq. (GHz)	Size (mm^2^)	Method Used	Return Loss (dB)	Amplitude (dB)	Phase Error
[34]	1.32 *λ*_0_ × 3.16*λ*_0_	79	5 × 12	Finline	<−10	−10 dB and −8 dB	16°
[35]	0.17*λ*_0_ × 0.23*λ*_0_	28	1.8 × 2.5	CMOS	<−10	−11 dB	22.5°
[36]	3.5*λ*_0_ × 4.3*λ*_0_	29	36.2 × 44.3	Substrate Integrated Waveguide	<−14	−9.8 dB	±6°~±9°
[37]	-	26–31.4		Shunt open, short stub		−15 ± 1 dB	±16°
[38]	5.76*λ*_0_ × 4.9*λ*_0_	20	86.40 × 74.19	Hollow waveguide	<−10	<1.8 dB	<19.6°
[39]	3.45*λ*_0_ × 4.7*λ*_0_	28	37 × 50	Switch IC in a single planar PCB	<−10	−2.9 dB	+7°
[40]	3.4*λ*_0_ × 3.4*λ*_0_	28	36.2 × 44.3	Planar design	<−10	5.3~ 10 dB	16°
This work	Internal: 2.17*λ*_0_ × 2.7*λ*_0_ Include extension line, Δ*L*: 9.34*λ*_0_ × 7.7*λ*_0_	28	Internal:23.26 mm × 28.92 mm. Include extension line, Δ*L*:100 mm × 82.5 mm	Via-hole with circular slot ground plane	<−10	Internal design: −6.1 dB to −8.1 dB Overall design: −8.6 dB to −10.3 dB	±30° with progressive phase difference

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
