# Peer review of "Insertion Loss and Phase Compensation Using a Circular Slot Via-Hole in a Compact 5G Millimeter Wave (mmWave) Butler Matrix at 28 GHz"

_sensors, 2022, doi:10.3390/s22051850_

Round 1

Reviewer 1 Report

This manuscript is on a compact dual-layer BM。The results presented here are quite interesting & valuable. The concept of the miniaturization design is new. I hope it can be accepted & published.

Author Response

Reviewer 1's Comment:

This manuscript is on a compact dual-layer BM. The results presented here are quite interesting & valuable. The concept of the miniaturization design is new. I hope it can be accepted & published.

Authors’ Response: Dear reviewer, Thanks for the valuable comments.

Reviewer 2 Report

The proposed work is, in my opinion, complete and solid.

For a better understanding of the material, I would suggest the Authors to address these MINOR points:

  1. Which is the dependance of the accuracy of the results by the mechanical tolerances of the manufactured structures as vias, via-holes, traces, etc.
  2. Which kind of boundary conditions have been used for the calculations reported in Fig. 5

The Authors are invited to add few comments.

Author Response

Reviewer 2’s Comments:

The proposed work is, in my opinion, complete and solid.

For a better understanding of the material, I would suggest the Authors to address these MINOR points:

The Authors are invited to add few comments.

Authors’ Response: Dear reviewer, Good day, Thanks for the comments and the review. Please find our response below:

1. Which is the dependance of the accuracy of the results by the mechanical tolerances of the manufactured structures as vias, via-holes, traces, etc.

Response: Mechanical tolerance has been looked through consistently while manufacturing the structure. However, there is one limitation of the thicknesses of the PC Clads which we use only the PC Clads available in the market. In this project, the structure of via hole elements of circular slot diameter, Ds, via- hole pin, DP and substrate thickness, DT affect the insertion loss and phase of the design structure as been investigated in Figure 8.

2. Which kind of boundary conditions have been used for the calculations reported in Fig. 5

Response: The boundary condition selected for Figure 5 is open space. However, we analyzed the electric boundaries as a reference.

Reviewer 3 Report

This paper proposes a compact dual-layer BM for 28 GHz mm-wave applications. The reviewer concerns and comments are listed below:

  1. The equation (13) is the same as equation (12). It should be port 3 in equation (13).
  2. In Figure 19, it should be phase output difference (Degree) instead of phase output (Degree). How the authors get the theory results? Please show the equations used for calculation.
  3. In Figures 24 & 25, the error between measurement and simulation are quite large. Please explain it.
  4. In Table 5, the amplitude of Ref.[40] is positive. Is it correct? The phase error of this work is quite larger than the references. Actually the size of this work is the largest one in Table 5. Why the feeding strip line extension is 30 mm?
  5. In Table 3, the unit of the parameters is missing.
  6. English needs to be improved since there are a lot of grammar errors.

Author Response

Reviewer 3’s Comment:

This paper proposes a compact dual-layer BM for 28 GHz mm-wave applications. The reviewer concerns and comments are listed below:

Authors’ Response: Dear reviewer, Good day, Thanks for the comments and the review. Please find our response below:

1. The equation (13) is the same as equation (12). It should be port 3 in equation (13).

Response: The equation (13) has been revised. Thanks for your comment.

2. In Figure 19, it should be phase output difference (Degree) instead of phase output (Degree). How the authors get the theory results? Please show the equations used for calculation.

Response: The label for the phase output difference has been revised to output phase difference. The output phase difference between radiating elements for a Butler matrix with N elements and for the pth beam location is given by Equation 9 from the Reference [31].

3. In Figures 24 & 25, the error between measurement and simulation are quite large. Please explain it.

Response: In Figures 24, the return loss measurement is lower than -10 dB. The frequency shifting might be attributed to the fabrication tolerance and variations in the high frequency connector properties. In Figure 25, The average error between simulation and measurement is 1.38 dB. This average error is considered acceptably small because it already includes the structure losses due to the additional stripline. We add additional stripline due to the structure is very small at the millimeter wave, c =fλ. This affect the final design of the Butler Matrix especially when the output is connected to the millimeterwave connector. In the design, there are four input connector and four output connectors, to get a stable structure and promising measurement, extension of stripline is used to satisfy the actual connector spacing. The simulation analysis of the stripline is performed in Figure 6, where this value satisfies the variation of the theoretical value, 6±3dB.

4. In Table 5, the amplitude of Ref.[40] is positive. Is it correct? The phase error of this work is quite larger than the references.

Response: Yes, it is correct. We found from the paper the amplitude for Ref [40] is positive.

Response: The phase error of this work is quite larger than the references because of the additional strip line due to the limitation in measurement. The actual performance of the phase error should be less. This refers to the analysis in Figure 6 for the possible phase error performance when the additional strip line is removed.

5. Actually the size of this work is the largest one in Table 5. Why the feeding strip line extension is 30 mm?

Response: The size of this work is the largest one in Table 5 because of the additional strip line is added due to the limitation in measurement. The actual size of the work itself will be smaller which is 23.26 mm × 28.92 mm (2.17λ0 × 2.7λ0). The additional stripline is attached to the final design by considering the correct amplitude and output phase of the internal design. The additional 30mm strip line to the BM is to satisfy the actual connector spacing.

6. In Table 3, the unit of the parameters is missing.

Response: The unit of the parameter has been added. Thanks.

7. English needs to be improved since there are a lot of grammar errors.

Response: The article has already been sent for proofreading. Thanks.

Round 2

Reviewer 3 Report

The reviewer concerns and comments are listed below:

  1. Please explain why there are some Chinese words 线性 in Figure 6 and Figure 8a,b c.
  2. Why are two curves in Fig. 8c so closed to each other? Please change the scale to make them separately.
  3. How can you get the conclusion that minimized insertion loss is obtained when DS is 1.2 mm, DP is 0.3 mm and DT is 0.254 mm?
  4. Please include the equation (Equation 9 [31]) used to obtain Figure 19 in the manuscript.
  5. Please include the discussion of the error between simulation and measurement of Figu. 24 in the manuscript.

Response to Reviewer 3: The phase difference between radiating elements for a Butler matrix with N elements and for the pth beam location is given by Equation 9 from the Reference [31].

Author Response

Dear Editor and Reviewers,

Thank you and we appreciate for your time in reviewing our paper and providing valuable comments.

Please find the response for all comments given below. All modifications in the manuscripts have been put in the tracking comments in word version also (Supplementary version). We have carefully considered the comments and we have tried our best to address every comment into our revised paper.

We hope the revised manuscript meets the standard of this journal. We welcome further constructive comments if any.

We have also sent the manuscript to a professional Language Editing Service.

Thank you and best regards,

Zubaida Yusoff, PhD

zubaida@mmu.edu.my

Multimedia University, Malaysia

The reviewer concerns and comments are listed below:

1. Please explain why there are some Chinese words 线性 in Figure 6 and Figure 8a,b c.

Response: Dear reviewer, no Chinese words can be viewed from our part.

2. Why are two curves in Fig. 8c so closed to each other? Please change the scale to make them separately.

Response: The scale has been changed accordingly. Thanks.

3. How can you get the conclusion that minimized insertion loss is obtained when DS is 1.2 mm, DP is 0.3 mm and DT is 0.254 mm?

Response: The initial analysis with the different circular slot diameter, DS, the via-hole pin diameter, DP, and substrate thickness, DT, towards the insertion loss is performed while having the other two parameters are constant respectively. The parameter results are explained in Figure 8. The minimized insertion loss is obtained from Figure 8 (a), where the insertion loss is -0.49 dB when the circular slot diameter, Ds = 1.2 mm. The variation of microstrip via-hole diameter, Dp in Figure 8 (b) shows the insertion loss of -0.53 dB when Dp = 0.3 mm and in Figure 8 (c), the insertion loss for the substrate thickness of DT =0.254 mm is -0.5 dB. The optimized parameter in Figure 8 is used during the final design structure of BM, and produce the minimized insertion losses to the final performance.

4. Please include the equation (Equation 9 [31]) used to obtain Figure 19 in the manuscript.

Response: The equation (Equation 9 [31]) can be found between row 236 and 237 in Section 3 Butler Matrix (Page 9).

5. Please include the discussion of the error between simulation and measurement of Figure 24 in the manuscript.

Response: The discussion of the error between simulation and measurement of Figure 24 has been included in the manuscript.
